# Simulating the Rapid Devolatilization of Mineral-Free Lignins

**DOI:** 10.3390/polym15204043

**Published:** 2023-10-10

**Authors:** Stephen Niksa

**Affiliations:** Niksa Energy Associates LLC, Belmont, CA 94306, USA; niksaenergy@gmail.com

**Keywords:** lignin, devolatilization, pyrolysis, reaction mechanism, FLASHCHAIN^®^, biomass

## Abstract

*Lig*-FC is a network depolymerization mechanism for the rapid primary devolatilization of mineral-free lignins that has already been validated with test data on 16 lignin samples. This paper expands the validation with an additional 13 lignins, including cases that applied different lignin preparations to the same feedstock. The validations reported here cover 27 mineral-free lignins for temperatures to 1150 °C, heating rates from 30 to 8000 °C/s, contact times after heatup to 90 s, and pressures from vacuum to 0.13 MPa. *Lig*-FC accurately depicts the impacts of lignin quality, heating rate, temperature, contact time, and pressure on the major products and oils’ molecular weight distributions (MWDs). All raw lignins contain abundant oil precursors that are released as oils via flash distillation as soon as a flow of noncondensables carries them into the free stream. Consequently, lignin MWD is an essential aspect of lignin constitution because it determines the inventory of inherent volatile chains subject to unhindered flash distillation. Lighter lignin MWDs have larger inherent inventories and therefore produce more oils than heavier MWDs at the onset of devolatilization. Oil yields diminish and char yields increase for progressively heavier MWDs and heavier mean monomer weights and for lignins with relatively less H and more O compared to C.

## 1. Introduction

This paper expands the validation of a reaction mechanism for the devolatilization of mineral-free lignins called “*lig*-FC”. The mechanism’s ultimate application is to accurately describe the release of gaseous fuels and chemical intermediates from the lignin component in any biomass form during suspension firing in pyrolyzers, furnaces, and gasifiers, and in the fluidized suspensions in atmospheric and circulating fluidized bed combustors, and fluidized bed pyrolyzers, and gasifiers. *Lig*-FC’s formulation is presented in the Supplemental Material to [1] along with its initial validation with 16 lignins. This paper expands the validation with an additional 13 lignins, including cases that applied different lignin preparations to the same feedstocks. The interpretations of the new validations are as accurate as that in the first validation series for the same ranges of model parameters. This work thereby demonstrates the accuracy of *lig*-FC over the full domain of mineral-free lignin characteristics, including any of the major preparation processes, any lignin MWD, and the full range of lignin compositions. However, it does not modify the theory or any of the detailed interpretations of the validation database in [1]. Consequently, the overview of *lig*-FC appears in [1], and the ultimate conclusions of this work are essentially the same as those in [1]. Readers unfamiliar with lignin and its devolatilization behavior and with the author’s modeling approach and reaction mechanisms will need to read [1] to comprehend what this paper is about.

The noncatalytic, spontaneous channels in *lig*-FC were initially formulated as part of a two-component devolatilization mechanism for whole biomass called “*bio*-FLASHCHAIN^®^” [2]. This first version of *lig*-FC was recently used to demonstrate an essential role for the flash distillation of oils during the devolatilization at atmospheric pressure of 14 lignins representing a broad range of lignin MWDs and chemical compositions [3]. The present study and its companion [1] update *lig*-FC with an expanded submodel for lignin constitution and additional structural features and reaction processes and validate simulation results with a much more extensive database of test results that covers most of the commercial operating domain.

The next section briefly Introduces the vocabulary for the main elements of the theory along with essential aspects of the lignin constitution submodel; much more detail is given in [1]. After the validation database and the strategy behind the kinetic parameter assignments have been presented, comparisons among simulation results and measured product distributions and MWDs illustrate the quantitative performance with 27 mineral-free lignins for temperatures to 1150 °C, heating rates from 30 to 8000 °C/s, contact times after heatup to 90 s, and pressures from vacuum to 0.13 mPa.

## 2. Overview of *Lig*-FC

According to *lig*-FC, the essential aspects of lignin devolatilization are: (1) reactions that randomly rupture monomers and thereby depolymerize lignin macromolecules over a broad fragment distribution; (2) bimolecular recombination reactions that coalesce the shortest, most volatile macromolecules into longer macromolecules; (3) fragmentation statistics to describe the probability that given monomer ruptures and recombinations produce fragments of a certain size; (4) a cascade of charring reactions that inhibit subsequent chain ruptures while bound monomers decompose sequentially into semichar units, char units, and annealed char plus noncondensable gases; and (5) the flash distillation analogy (FDA) whereby light fragments in the condensed phase vaporize into the escaping noncondensable gases, depending on the molecular weight, temperature, and pressure [4]. The analysis predicts the yields of CO, CO_2_, light oxygenated gases (OXYs), H_2_O, and H_2_; the yields and elemental compositions of the solid residue and bio-oil; and the MWD of bio-oil.

*Lig*-FC describes only primary lignin devolatilization and omits all secondary processes. Primary devolatilization exclusively entails chemistry in only the condensed solid phase, whereas secondary chemistry comprises the succeeding chemistry among the gaseous reaction products of primary devolatilization. *Lig*-FC would normally be coupled with distinct reaction mechanisms for conversion of both volatiles and char beyond the suspended biomass particles in simulations of any biomass utilization technology. In testing, secondary chemistry is avoided by imposing heating rates at least as fast as 5–10 °C/s and by sweeping the primary products away from hot surfaces in an inert gas stream.

In *lig*-FC, a rudimentary model of the initial chain distribution is assigned by fitting the proportions of reactant, intermediate, and volatile chains to either the degree of polymerization (DP) or to number-average (M_n_) or weight-average (M_w_) lignin molecular weights. As developed elsewhere [5], the simplest form for the distribution of chains in lignin is based on random scissions among nominally infinite chains and is given by:x_j_ = p(0)^(j − 1)^ (1 − p(0))^2^
(1)
where x_j_ is the number of j-mers divided by the initial number of monomers and p(0) is the probability for intact monomers in raw lignin. With this function, a lignin j-mer is depicted as a sequence of j − 1 intact monomers with a broken half-unit on each end. Marathe et al. [6] reported M_w_ for 14 lignins that were first used to specify p(0), which in turn specified the initial lignin MWD from Equation (1). These lignin MWDs are compared in Figure 1 to reported MWDs for two whole lignins and their light and heavy fractions; these two lignin triplets are among the heavier and lighter samples in this dataset. On average, the reported MWDs represent 91.4 ± 5.5 wt.% of the samples. The estimated MWDs are accurate for the lightest and heaviest portions of the MWDs but underestimate contributions for intermediate weights. The maxima in the simulated MWDs are accurate for most but not all these samples. 

The four variable aspects of lignin constitution in *lig*-FC are the lignin MWD (specified from DP, M_n_, or M_w_), monomer compositions from an ultimate analysis, mean monomer weights, and ash contents and compositions (which are characterized separately [7]). Table 1 shows the assigned compositions for the three structural units, mean monomer weights, and M_w_-values for the 27 lignins in the validation database. For the 14 lignins tested by Marathe et al. [6], the compositions were reported for only the whole lignin in each of the four triplets. Since the charring cascade preferentially eliminates O and H as noncondensables, all three element numbers diminish from monomers to semichar to char, where the char unit compositions for nearly all lignins saturate to 89 wt.% C, 4.2% H, and 6.8% O. The range of monomer weights from 208 to 237 g/mol is definitely large enough to affect oil production. The M_w_-values better express the extents of lignin MWDs and vary from 585 to 6170 g/mol. Those for the last nine samples in Table 1 were estimated as the values in Marathe et al.’s dataset for the same lignin type.

The validation database contains six laboratory studies that collectively determine complete product distributions including oil MWDs, including the study featured in the original validation work because it includes M_w_-values for the 14 tested lignins [6]. Marathe et al. [6] used a wire mesh reactor (WMR) to monitor the yields of oils, char, and gas for primary devolatilization at 5000 °C/s to 530 °C with a 5 s isothermal reaction period (IRP) under vacuum and at atmospheric pressure. Fourteen lignins were tested at these conditions, and one was also tested at four temperatures at both pressures. Essential aspects of the test procedures and supporting analyses were reported in [1].

Chen et al. [8] extracted one whole kraft lignin with an M_w_ of 4660 g/mol into three fractions with M_w_ of 820, 2390, and 6160 g/mol. Tests monitored the partitioning of these fractions into char and a partial gas mixture of CO_2_, CO, light gaseous hydrocarbons (GHCs), and H_2_. The differences to close mass balances were reported as “oils” when they actually are sums of oils, H_2_O, and OXYs. Samples of 100 mg were heated in a quartz U-tube that was inserted into a tube furnace at 400, 500, and 600 °C for 90 s. Reactor dimensions were not reported, so the heating rate cannot be estimated; the simulations are based on 100 °C/s. Moreover, the reported char yields at 400 °C vary from 0 to 78.8 wt.% across the four fractions, and three of four are lower than the char yields for 500 °C; both features are impossible. There are no irregularities in the reported yields for 500 and 600 °C, so only the dataset for 400 °C is ignored. The reported ultimate analyses for the four lignins are within 1.5 daf wt.% for C and O, so the simulations have the same parameter set for these four lignins. 

Nunn et al. [9] used a WMR to generate a few complete and mostly partial product distributions from one lignin for 1000 °C/s to 310–1170 °C with no IRP at 0.13 MPa. Collectively, the test records cover H_2_O, CO, CO_2_, OXYs, H_2_, and GHCs across the domain of operating conditions, plus a few elemental compositions for oils and char. Mass balance closures on individual tests are generally within ±5% across the entire database, and in the three tests with oil and char compositions, C/H/O balances were closed to within ±4%. The study’s greatest limitation is that secondary volatiles pyrolysis was unregulated, as there was no sweep stream to remove oils from the reactor hot zone before they could decompose or any other means to quench the volatiles as soon as they escaped from the lignin. The reported product distributions clearly display attributes of secondary pyrolysis, which become more pronounced for progressively hotter temperatures. In the validation work, the threshold temperature for negligible oil decomposition during heatup at 1000 °C/s is estimated as 750 °C, whereas all char yields were qualified. 

Thermal histories in these simulations are based on the reported heating rate and reaction temperatures. However, the mesh and sample were not force-quenched; rather, they cooled convectively at approximately −200 °C/s. Preliminary calculations determined that adding a 150 ms IRP for tests that imposed immediate cooling after heatup gave comparable incremental yields after the end of the heating period. In all simulations of this dataset, this 150 ms IRP was added to the reported IRPs in the thermal histories.

Zhou et al. [10] used a WMR to monitor gas, oil, and char yields from one lignin for 8000 °C/s to 365–685 °C with 3 s IRP under vacuum. They also monitored the MWDs of lignin and oils, albeit without a quantitative calibration, so these MWDs will not be considered. The mass balances closed to within ±6 wt.% but only for temperatures cooler than 500 °C, which is attributed to oil decomposition at hotter temperatures. Pecha et al. [11] used a modified pyroprobe reactor to determine char yields for 60 s at 500 °C under pressures from 0.01 to 0.10 MPa. Across this pressure range, the heating rates varied from 30 to 80 °C/s.

Custodis et al. [12] used a pyroprobe to heat six mineral-free lignins at roughly 10^4^ °C/s to 350–750 °C with 20 s IRP. It is not clear if volatiles were swept out of the hot zone throughout the test although there are no clear indications that oils decomposed at the hottest test temperatures. Tests monitored the partitioning of the lignins into char and a partial gas mixture of CO_2_, CO, and GHCs. The differences to close mass balances were denoted as “oils plus H_2_O” when they actually are sums of oils, H_2_O, H_2_, and OXYs. The most distinctive feature of this dataset is that the suite of lignins represented three lignin preparations applied to the same hardwood and softwood samples. Both woods were first extracted in a toluene/ethanol mixture, then the lignins were collected with treatments of dioxane/HCl, Klason hydrolysis, and organosolv in ethanol/H_2_SO_4_. Since the reported ultimate analyses for each lignin pair from the same recovery process were substantially different, distinctive parameter sets were assigned for each sample to fit the reported partitioning of lignins into char and aggregate oils. No lignin MWD information was reported, so the M_w_-values were estimated as 2000 and 2500 g/mol for softwood and hardwood lignins, respectively, except that the Klason hardwood lignin has an M_w_ of 3700 g/mol.

These six datasets are complementary: Marathe et al. thoroughly characterized oil MWDs from 14 lignins and Nunn et al. thoroughly characterized all the major product yields and compositions, albeit for a single lignin. Chen et al. expanded the range of lignin molecular weight. The tests from Pecha et al. and Zhou et al. substantially extend the range of heating rates in the validation database. Custodis et al. characterized the impacts of different lignin recovery processing on hardwood and softwood. All samples in this database have less than 0.1 wt.% ash and are regarded as mineral-free. Two other datasets qualified for lignin primary devolatilization are interpreted elsewhere with the same parameter ranges used here [1].

The kinetic parameters assigned for monomer rupture, recombination, monomer decomposition, semichar decomposition, and annealing are reported in [1], as are the three parameters in the saturated vapor pressure of oil precursors. Each test in the validation database was simulated with the reported heating rate, reaction temperature, and IRP. The simulations used reported test pressures except for runs under vacuum, which were simulated with a pressure of 0.01 MPa. Each simulation took less than a second on a desktop microprocessor.

## 3. Results

Since parameters were adjusted to match reported yields for every distinctive lignin composition in the validation database, the model results will be called “simulation results” rather than “predictions”. Figure 2 introduces the impact of the most important aspect of lignin quality, the lignin molecular weight. The data from Chen et al. [8] show the lignin partitioning into char, a partial gas mixture of CO_2_, CO, GHCs, and H_2_, and a mixture of oils, H_2_O, and OXYs assigned to close the mass balances. The simulated products are grouped in the same ways, except that the right panel also shows just the simulated oil yields for comparison. Since the reported ultimate analyses for the whole lignin and the three fractions are nearly the same, the simulations used a single parameter set for the four samples. At both temperatures, char yields increase and condensable yields decrease for progressively heavier lignins while the partial gas yields are hardly perturbed. Char yields diminish and partial gas yields increase by the same amounts for the hotter temperature, which reflects char decomposition and annealing. Normally, 600 °C is too cool for annealing, but due to the extended IRPs in these tests, it does make an appreciable contribution at the hottest temperature. The simulated condensable yields for both temperatures are indistinguishable because oil production finishes by the time the lignins reached 500 °C for this heating rate of 100 °C/s. Similarly, the measured condensable yields are the same within the measurement uncertainties. The simulated oil yields are somewhat less sensitive to lignin molecular weight than the condensable yields but only for weights heavier than 4500 g/mol. This transition occurs at the weight that has relatively few inherent oil precursors in the raw lignin, so that monomers must rupture to depolymerize the intermediate chains into the volatile size range before oils can be produced. 

Figure 3 shows the partitioning of lignin into oils and char for 5000 °C/s to 530 °C with 5 s IRP under vacuum and at 0.1 MPa for 14 lignins from the original validation work [1]. Here, the 14 samples are arranged by the sample number on the lower x- axis, and the M_W_-values for all samples appear across the top of the figure. Note that the M_W_-values are for discrete sample numbers and do not determine a continuous variation in this parameter along an axis. Each of the four triplets are arranged as whole, light, and heavy samples from left to right. Yields were not reported for sample 9 under vacuum and for samples 6 and 7 at 0.1 MPa. The prominent impact of lignin MWD is apparent in two ways in Figure 3: First, across each triplet, oil yields first increase for the lightest fraction, then decrease for the heaviest fraction, while the char yields change in the opposite ways. Second, except for samples 13 and 14, the MWDs of the whole lignins become lighter for progressively greater sample numbers. The simulation results depict both trends within the breaches in the mass balances for both pressures. In every instance, the simulated oil yields across each triplet increase, then decrease for whole, light, and heavy samples and vice versa for char yields. The measured yields at both pressures display this triplet tendency except for the char yield for sample 10 at 0.1 MPa and for oil yields for samples 7 or 8 and 12 under vacuum and for sample 10 at 0.1 MPa. The accurate triplet tendency from *lig*-FC is especially significant because it is evident in the measured char yields for all four triplets, despite large variations in composition among these four samples (cf. Table 1), and also because the cases in each triplet were simulated with the same parameter set. The impact of lignin MWD variations among only the samples with different compositions is seen in the cases for the whole lignins, which are the first in each triplet for samples 1–12 and also samples 13 and 14. Simulated oil yields increase and char yields diminish for progressively heavier lignin MWDs at both pressures. The measured yields abide by this tendency except for samples 10 and 14 at both pressures. None of these small deviations are important because the simulated lignin partitioning is quantitively accurate in every test case. Oil yields diminish and char yields expand across the range of M_W_ because heavier lignin MWDs have fewer inherent oil precursors to be quickly swept away at the onset of devolatilization. 

The other striking feature in Figure 3 is the reduction in oil yields by almost half between vacuum and 0.1 MPa. This feature and the way that oil yields become insensitive to temperature at 0.1 MPa (seen in [1]) could only be depicted when bimolecular recombination was included in *lig*-FC. Without recombination, the simulated differences in the oil yields were only half as large as the measured differences. 

Figure 4 compares simulated and measured oil MWDs for the same test conditions as Figure 3 with the two lignin triplets, whose raw MWDs appear in Figure 1. As seen in Figure 1, the triplet MWDs denoted as 1/2/3 extend to much heavier weights than those for triplet 7/8/9. Each case has the oil MWDs from the whole lignin plus its light and heavy fractions. The reported and simulated oil MWDs pass through maxima around 500 g/mol, then gradually relax in long tails extending to much heavier molecular weights, as in gamma distribution functions. The simulated MWDs are shifted toward heavier weights for the heavier lignin in each triplet, as seen in the measured MWDs. The simulated MWDs are particularly accurate for the oils from triplet 7/8/9 at both test pressures. Those from triplet 1/2/3 overestimate the maxima under vacuum and underestimate the maxima at atmospheric pressure. Some of the discrepancies should be attributed to a deliberate truncation of the simulated oil MWDs at 2000 g/mol. This author is skeptical that the MWDs of vaporized organic oils can extend to 5000 g/mol and suspects that non-exclusion effects in GPC for highly polar macromolecules such as bio-oils may be responsible for the extended tails. This truncation necessarily steepens the lighter portion of the vacuum oil MWDs from triplet 1/2/3. However, the excessive tail is absent from the reported vacuum MWDs from the lighter triplet and from both triplets at 0.1 MPa, for which the MWDs are accurate across the entire lignin weight range. Overestimates for the increments of intermediate molecular weights in the lignin MWDs in Figure 1 do not carry over to the oil MWDs, because the FDA introduces independent dependences on oil weight, temperature, and pressure.

Number-average molecular weights for the 14 lignins in Marathe et al.’s dataset are evaluated for both test pressures in Figure 5. The simulated oil weights are accurate at atmospheric pressure but the truncation of the vacuum MWDs described above is responsible for the underprediction for the heaviest vacuum oils. Notwithstanding, the simulated M_n_ under vacuum of oils increases for progressively hotter temperatures in close agreement with the reported temperature dependence, which corroborates the temperature dependence in the FDA. The simulated temperature dependence is weaker for atmospheric oils but does not vanish like it does in the reported M_n_ values. According to the FDA, oil MWDs become lighter at progressively higher pressures because the longest volatile chains no longer have sufficient volatility to vaporize from the condensed phase (because the oil production rate is proportional to the ratio of the vapor pressure of each chain size to the ambient pressure [1]). Consequently, the cumulative oil sample becomes enriched in the lightest oil chains while the longer chains remain in the condensed phase where they are susceptible to recombinations that make them longer and therefore even less volatile.

The temperature dependence of oil and char yields are evaluated in Figure 6 for temperatures from 365 to 685 °C with very rapid heating and 3 s IRP under vacuum. Even at 365 °C, the reported partitioning is nearly complete, so that both the measured and simulated oil yields increase by only 20 wt.% for progressively hotter temperatures to 500 °C and then saturate to an ultimate yield for hotter temperatures. The reported and simulated char yields diminish by almost 15% over this temperature range. *Lig*-FC attributes the large oil yields at low temperatures to the abundance of inherent volatile chains in this lignin, whose estimated M_w_ is just under 1400 g/mol for which over 80% of the lignin is in the volatile size class. The modest temperature dependence through 500 °C reflects the relatively low activation energies for monomer rupture and decomposition, whereas the saturation in the oil yields for hotter temperatures reflects the rapid acceleration in the recombination rate due to its high activation energy.

Figure 6 also evaluates lignin partitioning for pressures from 0.01 to 0.10 MPa. Both the measured and reported char yields nearly double across this pressure range, which again corroborates that oil production rates are proportional to the ratio of the vapor pressure of each chain size to the ambient pressure, and chains that do not vaporize are ultimately incorporated into char. 

The complete distribution of products reported by Nunn et al. [9] is the basis for the evaluations in Figure 7 and Figure 8. The simulated lignin partitioning into oils and char in Figure 7 is generally within the measurement uncertainties, and the oil yields are especially accurate. For this heating rate, oil production begins at 300 °C; reaches its maximum rate at 450 °C; and finishes at 600 °C. These temperature windows shift toward hotter temperatures for progressively faster heating rates (as seen below). Simulated yields of four of the five major noncondensables are evaluated in Figure 8; Nunn et al. did not monitor H_2_. *Lig*-FC correctly predicts that OXYs and CO_2_ are released first while CO is generated on a longer time scale. The reported H_2_O yields exhibit hardly any temperature sensitivity, which is both implausible and also seen in the dataset on hemicellulose from this laboratory [13]. The simulated moisture yields evolve in tandem with CO. The ultimate yields of all these products are within the measurement uncertainties. The agreement represents adjustments to 12 of the 14 stoichiometric coefficients for gas production in *lig*-FC (excluding annealing) and is neither statistically significant nor unique, because a single dataset cannot determine so many parameters with stringency. The simulated dynamics of gas release are consistently faster than in the reported gas release histories and shift toward cooler temperatures by about 50 °C. This flaw is inconsequential because it was also seen in the cellulose devolatilization histories from this older pyrolysis system [5,13] compared to recent histories. 

The simulated H_2_ yield is 1.2 wt.% at 750 °C, whereas H_2_ was not measured. It is intended to represent both molecular H_2_ and the hydrogen in the reported ultimate yields of 0.8 wt.% CH_4_ and 0.5% C_2_ GHCs, because GHCs are omitted from the noncondensables in *lig*-FC. It exceeds the hydrogen in the reported GHCs by slightly less than 1 wt.%. 

The simulated C/H/O compositions of oils and char are evaluated in Table 2. The measured oil composition for 515 °C was estimated by closing the element balances with the reported yield and composition for char and the yields of all gases. The composition for 700 °C omitted an O-content, which was also estimated. Further, sums of all the measured C/H/O compositions breached the mass balance by up to 10%, which was eliminated with normalization. Simulated oil compositions for 500 and 515 °C are essentially the same, whereas the measured composition for 500 °C is substantially different than the estimated composition for 515 °C. The simulated oil composition matches the estimated composition but has more C and less O than the measured composition. Conversely, the simulated oil composition for 700 °C has less C and more O than the measured composition but exhibits the correct temperature dependence. The simulated char composition for 310 °C reproduces the lignin composition, which this measured composition replicates within 2 wt.% on C and O. The simulated char composition for 515 °C has more C and H and less O than the measured composition, which reflects the faster gas release rates in the simulations that gave greater yields of all gases except H_2_O than the measured values at this temperature (in Figure 8). Although the simulated compositions for oils and char are reasonably accurate, the evaluation of the oil composition is ambiguous because the only other comparable dataset [14] gives an oil H-content that is almost 40% greater than the lignin H-content, whereas Nunn et al. gives two oils’ H-contents that are slightly lower than the lignin H-contents. This discrepancy in the measurements can only be resolved with additional oil compositions for diverse lignins that close element balances in individual tests. 

The impact of the lignin recovery process is characterized in Figure 9 for lignins recovered from hardwood and softwood with dioxane, Klason, and organosolv treatments. Char yields were monitored directly, whereas the aggregate yields of oils/H_2_O/H_2_/OXYs were assigned to close mass balances. Since the parameter sets were tuned-in to these datasets, there are minimal discrepancies in the simulated results across the entire sample suite. The simulated and measured partitioning of lignins into char and aggregate oils for both woods treated with dioxane is indistinguishable across the entire temperature range. These lignins are the most reactive, by far; their char yields reached 50 wt.% at 470 °C. Both the Klason and organosolv lignin pairs have appreciably different thermal responses between the hardwood and softwood samples. For the Klason lignins, the partitioning through 550 °C is very similar. However, for hotter temperatures, the Klason hardwood lignin produces hardly any additional aggregate oils, while this softwood lignin almost doubles its oil production to an ultimate aggregate oil yield approaching 70 wt.%. For the organosolv pair, the partitioning of the softwood lignin is substantially weaker across the entire temperature range, so that the ultimate aggregate liquid yield is lower by 10 wt.% even though the ultimate char yields are comparable. Organosolv lignins also exhibit the slowest devolatilization rates of all samples in this suite, reaching 50% char yields from 550 to 600 °C. 

Klason treatment promotes the partitioning of softwood vs. hardwood, opposite to the slower devolatilization of organosolv softwood. Dioxane treatment gives the same devolatilization behavior for lignins from hardwood and softwood and also the fastest devolatilization rates among the three treatments. Although *lig*-FC can depict these influences across a broad temperature range, it does not interpret them because there are no firm tendencies among the assigned parameter sets for the various treatments and wood grades. 

## 4. Discussion

*Lig*-FC depicts all the essential aspects of lignin devolatilization for the target applications, and this study broadens the validation of the following tendencies: (i) oil yields diminish for progressively heavier lignin MWDs while char yields increase; (ii) oil MWDs shift toward heavier weights for progressively heavier lignin MWDs; (iii) oil yields are markedly greater under vacuum than at atmospheric pressure; (iv) oil MWDs become heavier for progressively lower pressures and hotter temperatures; and (v) oil yields increase for progressively hotter temperatures, especially under vacuum. Since lignins in each triplet in Marathe et al.’s dataset and the four lignins tested by Chen et al. have the same compositions, only seven parameter sets accurately interpreted the partitioning of 18 distinctive lignins, and only the frequency factors for recombination and for the first two steps in the charring cascade were varied. These validations are especially stringent because they were supported by measured M_w_-values for all 18 lignins and collectively cover a broad domain of operating conditions. Only the impact of the lignin recovery process remains inconclusive. *Lig*-FC also accurately depicts complete distributions of the major noncondensables and reasonable elemental compositions of oils and char, although many more complete product distributions for diverse lignins need to be reported to finalize the stoichiometric coefficients.

The only variable operating condition not represented clearly in the validation database is heating rate, which was varied over a wide range but for different lignins, pressures, and contact times. The simulation results in Figure 10 isolate the impact of heating rate variations for Nunn et al.’s lignin [9] with no IRP at 0.13 MPa. Ultimate temperatures were increased with faster heating rates to ensure that each devolatilization history covers oil production in its entirety without additional gas release due to semichar decomposition and annealing on much longer time scales. Oil production shifts to hotter temperatures for progressively faster heating rates, as expected. Its temperature window grows from 200 °C for 10 °C/s to 370 °C for 10^4^ °C/s, while temperatures for the maximum production rates increase from 340 to 670 °C. The temperature shifts are responsible for the substantially heavier oils for progressively faster heating rates, indicated in Figure 10 by the M_n_-values for the ultimate cumulative oil sample, which surge for heating rates faster than 300 °C/s. Over this range of heating rate, ultimate oil yield increase from 26 to 62 wt.%, while char yields diminish from 37 to 15%. The gas yields fall in tandem with the char yields. Unfortunately, measured lignin partitioning across such a broad range of heating rate, with all else the same, has not yet been reported, although one evaluation reported elsewhere [1] corroborates these tendencies for rates of 60 and 6000 °C/s under vacuum.

*Lig*-FC depicts the distinctive devolatilization behavior of individual lignin samples in terms of the lignin MWD, the elemental compositions of monomers, the average monomer molecular weight, and mineral levels. Mineral catalysis was excluded from this study and characterized separately [7], whereas the other three factors are further characterized here and elsewhere [1]. The key to the impact of lignin MWD is the abundance of inherent oil precursors in all raw lignins, especially in lignins with M_w_-values lighter than 2500 g/mol that have half or more of their macromolecules in the volatile size range [1]. With so many oil precursors in the lignin, only a source of noncondensable gases is needed to produce oils, and monomer decomposition and rupture produce the first noncondensables. This aspect is isolated in Figure 11, which shows devolatilization histories for 5000 °C/s to 530 °C under vacuum and at 0.1 MPa for two lignins with the same compositions and monomer weights. Their only difference is that one has a MWD whose M_w_ is 1000 g/mol and the other has 3600 g/mol (the actual lignin MWD for Nunn et al.’s sample [9]). Since the compositions are the same, the same parameter set is used for both cases. To compare these cases, ignore the results under vacuum for the time being and focus on those at 0.1 MPa. The unconverted condensed phase is resolved into volatile and intermediate size ranges, and these portions clearly show which processes are responsible for the impact of lignin MWD because recombination is the only means in *lig*-FC to lengthen macromolecules. The light lignin essentially contains only volatile chains, whereas the heavy sample has one-third volatile chains and half intermediate chains, the remainder being reactant (not shown). With the light sample, the volatile chain inventory abruptly falls to 25 wt.% in only 150 ms. Such a small inventory suppresses bimolecular recombination, so that the intermediate inventory surges at 100 ms to only 10 wt.%, then gradually reaches 15% when all volatile chains have been converted. With the heavy sample, the volatile inventory first increases to 45 wt.% as intermediates are ruptured, then gradually diminishes through 600 ms. Since volatile chains persist at elevated levels, recombinations counteract the rupture of intermediates at 200 ms and then sustain the approach to the saturation level of 25 wt.% for the ultimate char yield. 

These differences in the MWDs of the condensed phase strongly affect oil production. With the light sample, oils are produced as soon as gases are released, first via monomer rupture and decomposition and then via recombination. From the onset of gas production at 75 ms, flash distillation releases oils from the near-complete inventory of volatile chains. By the time recombination begins at 100 ms, nearly half the ultimate oil yield has already been released. Thereafter, recombinations and the charring cascade compete more effectively with flash distillation to retard oil production and to grow the inventory of intermediate chains that comprise the ultimate char. The transition from unhindered flash distillation of inherent volatile chains to a competition among flash distillation, recombination, and the charring cascade is also evident with the heavier sample. However, unhindered flash distillation from 75 to 100 ms produces less than 10 wt.% oils vs. 30% with the light lignin, because the inherent volatile chain inventory is only one-third of the lignin vs. almost 100% with the light lignin. Thereafter, the competition among flash distillation, recombination, and charring drives oil production to the ultimate saturation level. However, it does not compensate for the different outcome of unhindered flash distillation, so the ultimate oil yield is only 56 wt.% vs. 70 wt.% for the light lignin. Lignin MWD is an essential aspect of lignin constitution because it determines the inventory of inherent volatile chains subject to unhindered flash distillation. Lighter lignin MWDs have larger inherent inventories and therefore produce more oils than heavier MWDs at the onset of devolatilization. These differences largely vanish during the succeeding competition among flash distillation, recombination, and the charring cascade.

The impact of variations in the monomer weight was assessed elsewhere [1] in simulations with lignin samples that had the same compositions and MWDs but monomer weights of 215 and 251 g/mol. For rapid extended heating at 530 °C at 0.1 MPa, the simulated char yield increased by 4.5 wt.%, while the oil yield decreased by 5.5 wt.% for the heavier monomer. The oils became heavier, so that M_n_ shifted from 480 to 510 g/mol and M_w_ shifted from 650 to 690 g/mol. The mean monomer weight is an essential aspect of lignin constitution because it directly factors into the instantaneous weight of the vaporizing volatile chains throughout devolatilization. Variations in lignin elemental composition were assessed elsewhere [1] with 16 samples with uniform monomer weight and lignin MWD but different compositions, which was expressed by distinctive kinetic parameter sets. 

Finally, one of the most striking features in the validation database is the impact of pressure on the reported oil yields in Figure 3. The behavior is interpreted in Figure 11, where the focus is now on the simulation results under vacuum and at atmospheric pressure with only the heavier lignin. Under vacuum, the extremely fast vaporization of inherent volatile chains rapidly depletes the volatile chain inventory, while monomer rupture depletes the intermediate inventory. Consequently, within 200 ms, both inventories have fallen to 10 wt.% or less. At such a low level, bimolecular recombination is slow so the intermediate inventory remains low. In contrast, at 0.1 MPa, the volatile chains have lower volatility, which allows their inventory to first expand to 45 wt.% via ruptures of reactant and intermediate chains and to then decay gradually through 600 ms. The recombination rate is accelerated by the greater inventory of volatile chains, which drives the saturation level for intermediate chains over 20 wt.%. The ultimate level is half the initial level. Suppressed volatility at atmospheric pressure in conjunction with faster recombination reduce the ultimate oil yield to 56 wt.% vs. almost 80 wt.% under vacuum. These disparities are even more pronounced at hotter temperatures because the relatively large activation energy assigned for recombination preferentially accelerates recombination at hotter temperatures compared to monomer rupture and decomposition.

## 5. Conclusions

The expanded validation cases in this study further buttress the conclusions in [1], which are:(1)*Lig*-FC identifies the essential reaction mechanisms in primary lignin devolatilization as depolymerization via monomer rupture in competition with charring via a spontaneous three-stage cascade and bimolecular recombination and flash distillation of oil precursors from the condensed phase.(2)In *lig*-FC, the first gases form by monomer decomposition and rupture. Flash distillation moves inherent oil precursors from the condensed phase into the stream of noncondensables, which carries them away as product oils. Hence, the induction of primary lignin devolatilization does not entail depolymerization; only monomer decomposition and flash distillation of the inherent oil precursors are required.(3)Lignin MWD determines the inventory of inherent volatile chains subject to unhindered flash distillation. Lighter lignin MWDs have larger inherent inventories and therefore produce more oils than heavier MWDs at the onset of devolatilization. Raw lignins with M_W_ up to 2500 g/mol have at least half their monomers in inherent oil precursors.(4)The distinctive devolatilization behavior of individual lignins can be depicted from the lignin MWD, the elemental compositions of monomers, and the average monomer weight. Oil yields diminish and char yields increase for progressively heavier MWDs and mean monomer weights and for lignins with relatively little H and abundant O compared to C.(5)Bimolecular recombination is required to interpret the disparate ultimate oil yields under vacuum and at 0.1 MPa. Suppressed volatility at atmospheric pressure in conjunction with faster recombination reduce the ultimate oil yields to only half of those under vacuum.

## Figures and Tables

**Figure 1 polymers-15-04043-f001:**
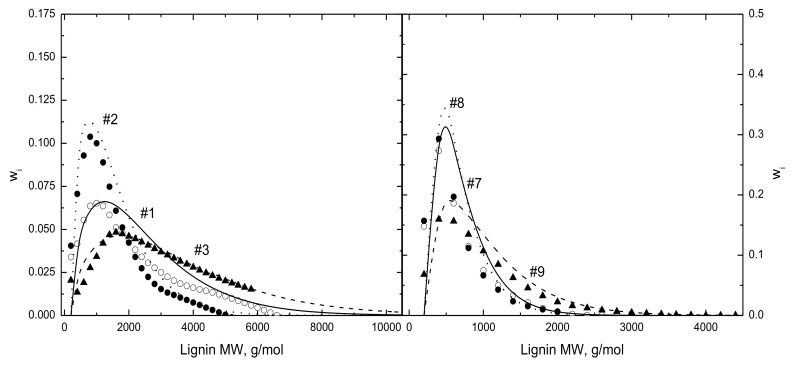
Comparisons of MWDs reported by Marathe et al. [6] for (◯) whole lignin and (●) light and (▲) heavy fractions to the MWDs given by Equation (1) that match reported M_w_-values for (**left**) heavy and (**right**) light lignin triplets, where the numbers for each triplet increase in order for whole lignin, light fraction, and heavy fraction.

**Figure 2 polymers-15-04043-f002:**
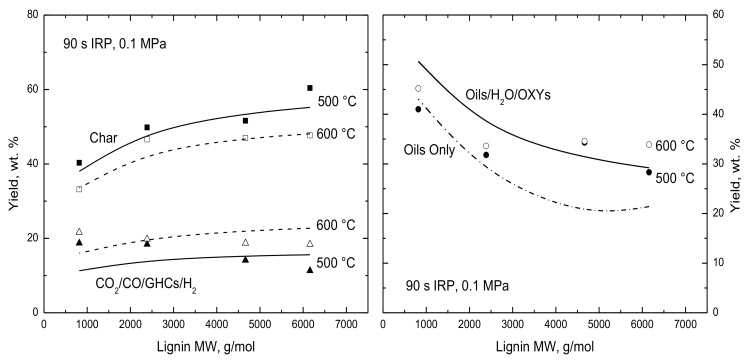
Evaluation of lignin partitioning vs. lignin molecular weight with (**left**) yields of (squares) char and (triangles) CO_2_/CO/GHCs/H_2_ from Chen et al. [8] for rapid heating to (filled symbols and solid curves) 500 and (open symbols and dashed curves) 600 °C with 90 s IRP and with (**right**) (circles and solid curve) oils/H_2_O/OXYs. The dot-dash curve shows the simulated oil yields only.

**Figure 3 polymers-15-04043-f003:**
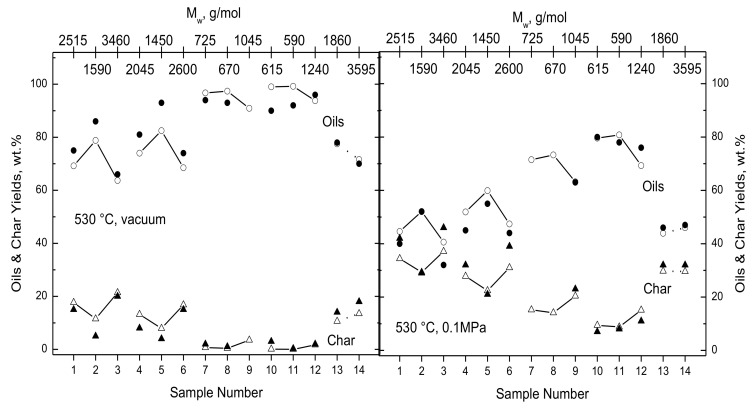
Evaluation of (open symbols connected by line segments) simulated lignin partitioning for 5000 °C/s with 5 s IRP at 530 °C with yields of (●) oils and (▲) char reported by Marathe et al. [6] under (**left**) vacuum and at (**right**) atmospheric pressure.

**Figure 4 polymers-15-04043-f004:**
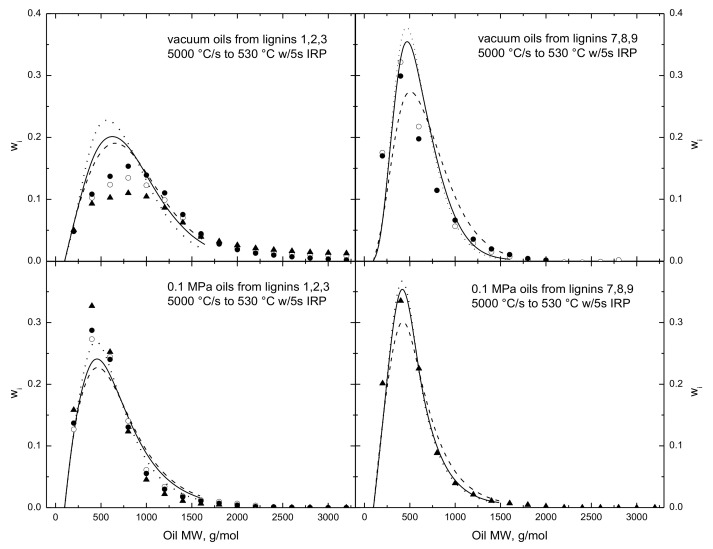
Evaluation of (curves) simulated oil MWDs for (left) a relatively heavy and (right) a relatively light lignin triplet, where each triplet contains (◯ and solid curves) the whole lignin, (● and dotted curves) a light fraction, and (▲ and dashed curves) a heavy fraction. The evaluations under (upper panels) vacuum and at (lower panels) 0.1 MPa are for 5000 °C/s to 530 °C with 5 s IRP.

**Figure 5 polymers-15-04043-f005:**
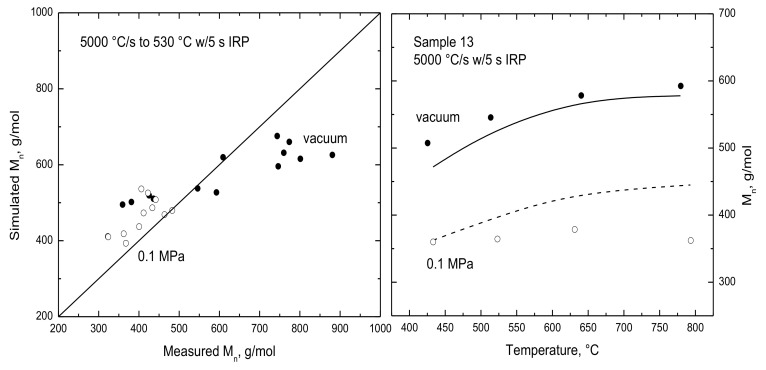
(**Left**) Parity plot of simulated vs. measured oil M_n_ under (●) vacuum and at (◯) 0.1 MPa, and (**right**) evaluation of M_n_ for sample 13 across a temperature range. Reproduced from [1] with permission from Elsevier, Copyright 2022.

**Figure 6 polymers-15-04043-f006:**
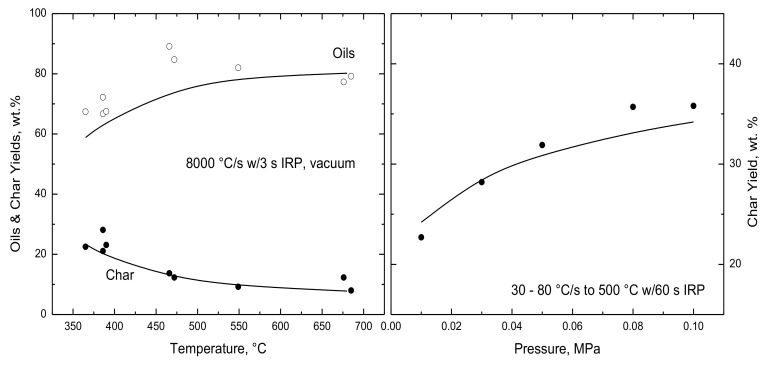
Evaluation of simulated lignin partitioning with measured yields of (◯) oils and (●) char from (**left**) Zhou et al. [10] for 8000 °C/s to various temperatures with 3 s IRP under vacuum and (**right**) Pecha et al. [11] for much slower heating to 500 °C with 60 s IRP across a pressure sweep.

**Figure 7 polymers-15-04043-f007:**
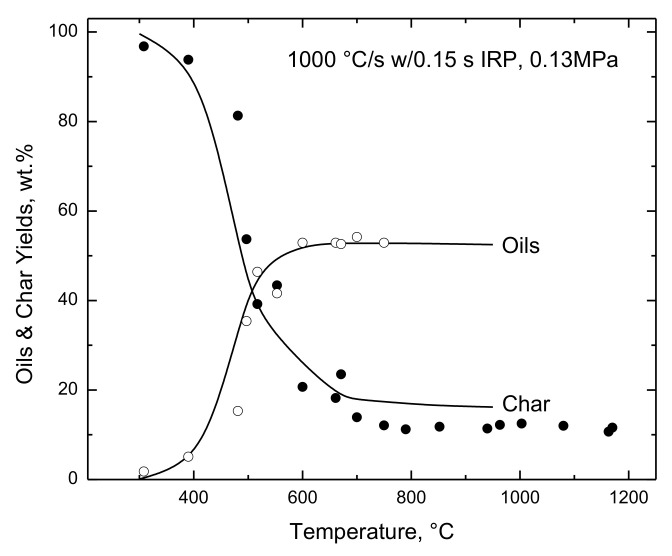
Evaluation of simulated lignin partitioning with measured yields of (◯) oils and (●) char from Nunn et al. [9] for 1000 °C/s to various temperatures with no IRP at 0.13 MPa.

**Figure 8 polymers-15-04043-f008:**
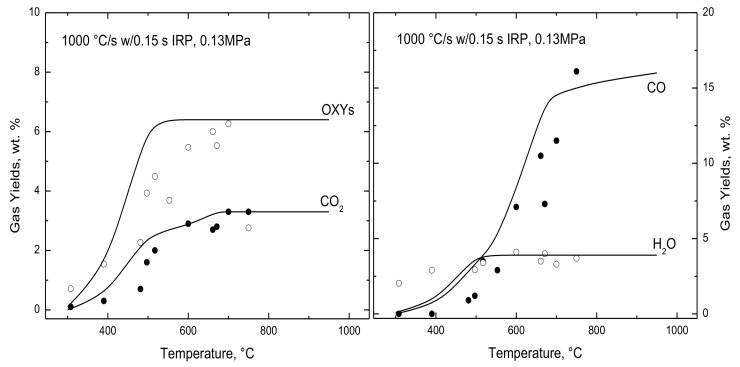
Evaluation of major noncondensable yields for the conditions in Figure 7.

**Figure 9 polymers-15-04043-f009:**
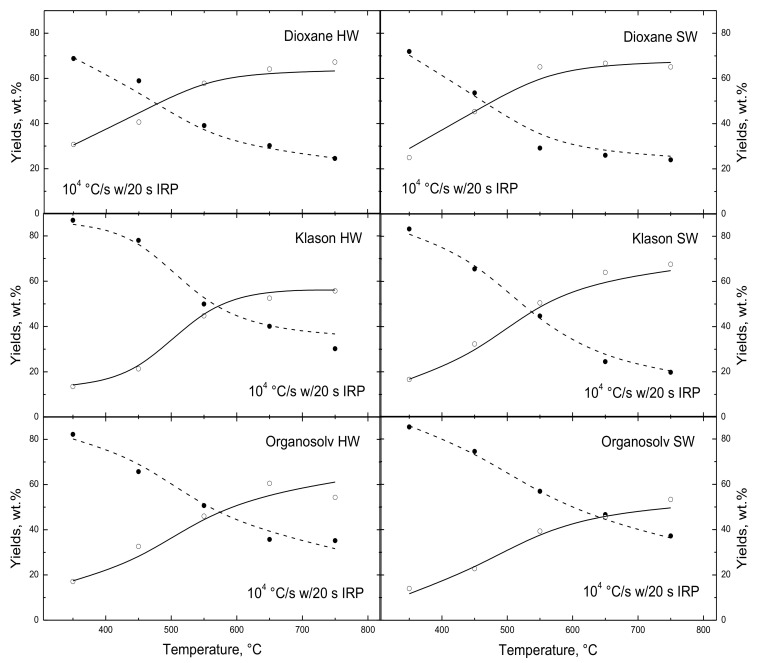
Evaluation of simulated yields of (dashed curves) char and (solid curves) oils/H_2_O/OXYs/H_2_ with measured (●) char and (◯) condensables yields from Custodis et al. [12] for rapid heating with 20 s IRP on lignins from three recovery processes applied to (**left**) hardwood and (**right**) softwood.

**Figure 10 polymers-15-04043-f010:**
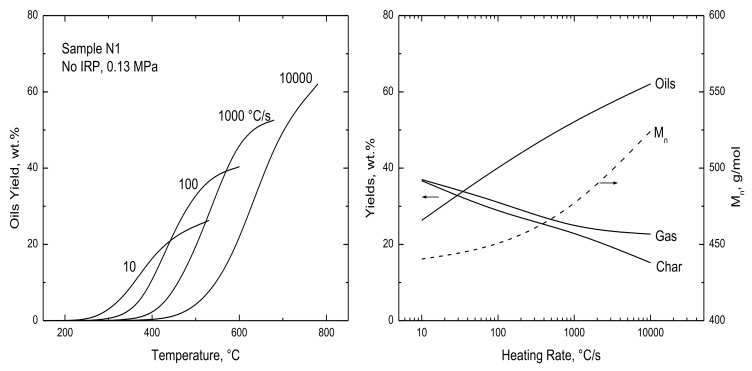
Simulated (**left**) oil production and (**right**) ultimate yields of (solid curves) gas, oils, and char and (dashed curve) oil M_n_-values for four heating rates with no IRPs at 0.13 MPa.

**Figure 11 polymers-15-04043-f011:**
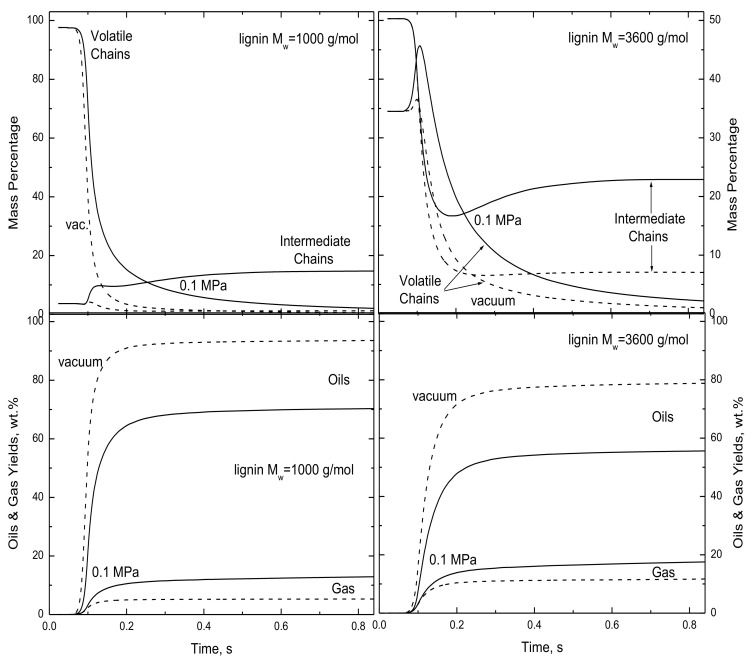
Simulated (**top**) char constitutions and (**bottom**) oil and gas yields for 5000 °C/s to 530 °C with 5 s IRP at (solid curves) 0.1 MPa and under (dashed curves) vacuum from lignins with M_w_-values of (**left**) 1000 and (**right**) 3600 g/mol.

**Table 1 polymers-15-04043-t001:** Assigned element numbers of structural units, monomer weights, and M_w_-values for lignins in the validation database.

Label	Monomer				Semichar			Char			M_w_
	C	H	O	<M_M_>	C	H	O	C	H	O	g/mol
M1 [6]	12.0	12.9	3.6	215	11.3	11.6	3.2	9.6	5.4	0.5	2515
M2											1586
M3											3457
M4	12.0	12.9	4.0	222	11.1	11.1	3.4	9.8	5.4	0.6	2042
M5											1441
M6											2596
M7	12.0	13.4	3.4	211	11.1	11.6	2.8	8.8	5.0	0.5	724
M8											675
M9											1046
M10	12.0	14.5	4.0	222	11.1	12.7	3.4	8.1	4.5	0.5	619
M11											585
M12											1247
M13	12.0	12.9	4.3	225	11.0	11.0	3.7	7.8	4.4	0.4	1862
M14	12.0	15.0	4.9	237	10.7	12.2	4.0	9.4	5.4	0.5	3604
C1 [8]	12.0	13.2	4.5	234	11.4	11.5	3.5	10.4	7.7	2.0	4660
C2											6170
C3											2380
C4											840
N1 [9]	12.0	14.7	5.0	236	10.2	11.5	2.8	7.9	6.6	0.4	^e^ 3609
P1 [10]	12.0	13.4	4.1	221	11.1	10.2	2.3	10.5	3.8	0.5	^e^ 2356
Z1 [11]	12.0	13.4	3.9	215	10.8	9.7	0.6	10.8	7.1	0.6	^e^ 1386
Cu1 [12]	12.0	15.0	4.9	235	10.7	11.0	3.6	10.2	1.4	1.5	^e^ 2050
Cu2	12.0	13.3	4.2	224	10.8	9.4	2.8	10.3	1.5	0.8	^e^ 2500
Cu3	12.0	13.2	4.6	231	10.9	10.9	3.9	10.9	0.3	0.8	^e^ 3770
Cu4	12.0	12.6	4.0	220	11.8	12.0	3.5	11.5	3.4	2.2	^e^ 2060
Cu5	12.0	11.0	3.7	213	11.8	10.3	3.2	11.5	1.7	1.9	^e^ 2000
Cu6	12.0	10.8	3.4	208	11.8	10.1	2.9	11.5	1.5	1.5	^e^ 2520

^e^ Estimated value.

**Table 2 polymers-15-04043-t002:** Evaluation of simulated elemental compositions of oils and char with data from Nunn et al. [9] for 1000 °C/s with no IRP at 0.13 MPa.

	T, °C	Measured	Simulated
Oils C/H/O	500	60.2/6.0/33.9	64.3/6.3/29.4
	515	64.9/6.0/29.4 *	64.1/6.3/29.2
	700	69.6/5.8/24.6 *	65.1/6.3/28.6
Char C/H/O	310	62.9/6.1/30.9	60.9/6.2/32.9
	515	64.3/5.7/30.0	68.3/6.4/25.3

* Estimated to close element balances.

## Data Availability

The author is not authorized to share test data that first appeared in the cited references.

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
