# Peer review of "Simulating the Rapid Devolatilization of Mineral-Free Lignins"

_polymers, 2023, doi:10.3390/polym15204043_

Round 1
Reviewer 1 Report
The manuscrint 2561780 “Simulating the rapid devolatilization of mineral‐free lignins” is described the hypothesis on mechanism of lignins fast pyrolysis.
Some remarks on the text are listed below.
Fig. 1. The title is not descried the right picture.
Table 1. Labels are not decoded, lignins are not described. Rows DP and MW are not connected to other ones. Are they described initial lignins or chars?
M1[6] - M1 [6] – a space.
Comparing DP and MW leads to the conclusion on monomers molecular weight: it is of 396-619 g/mol. For comparison, synapyl alcohol, the heaviest phenylpropane unit forming lignins is of 230 g/mol MW. What are monomers in the manuscript?
Why do you use C12 for monomers stoichiometry description? Syringyl FPU contains 11 C atoms only.
Line 201. andthat for monomer decomposition was raised to 113 kJ/mol because the sensitivity to
Table 2. The ratio of energy activation for monomer rupture (113 kJ/mol) and recombination (209 kJ/mol) is very strange: radical recombination occurs without activation barrier usually.
Line 234. GHCs, OXYs abbreviations are not decoded.
Fig 2, right. Fitted lines are not fitted the experimental points.
Line 561. weights of 215 and 251 g/mol. ??? For rapid extended heating at 530 °C at 0.1 MPa, the
Figure 3. Evaluation of (open symbols connected by line segments) simulated lignin partitioning for 5000 °C/s with 5s IRP at 530 °C with yields of () oils and (p) char reported by Marathe et al. [6] for (left) vacuum and (right) atmospheric pressure. What are empty symbols? I don’t understand, why abscissa is number of Lignins, but not Molecular weight?
Maximum sum of oil and char yields is of 95-97%. And what about H2O, CO2 CO formation (Fig. 8)?
Figure 4. Upper-left figure – fitting is not satisfied.
Figure 5. (Left) correlation is not satisfied. Right-lower fitting is not satisfied.
Table 3. The same remarks. Mesured oxygen content in oils is of 33.9 – 24.6%, decreasing by 9.3% abs and 27% relative. Simulated results differ only by 0.9% abs or 3% relative. Hence, the model do not predict correctly the obvious decrease of the oxygen content in oils while increasing temperature of pyrolysis. Prediction (simulation) of the char composition is also insufficient.
Simulating model is not described in the manuscript.
To conclude, the reviewer do not recommend the manuscript for publication, because the model are not described, and the hypothesis on the mechanism of pyrolysis is not efficient, it does not describe the results satisfactorily.

Author Response
Responses to Reviewer No. 1 for Polymers 2561780 by Stephen Niksa
The manuscrint 2561780 “Simulating the rapid devolatilization of mineral‐free lignins” is described the hypothesis on mechanism of lignins fast pyrolysis. Some remarks on the text are listed below.
Fig. 1. The title is not descried the right picture.
Response: Thanks. To the caption I added “…for (left) heavy and (right) light lignin triplets.”
Table 1. Labels are not decoded, lignins are not described. Rows DP and MW are not connected to other ones. Are they described initial lignins or chars?
Response: The lignins in Table 1 are denoted by the indices in the first column. For those from [6], I used the same numbering system as this testing team. The others can be cross-referenced with the properties in this table. I think the reviewer wants to know why the product of the monomer weight in the <MM> column and DP does not equal MW for the raw lignin. That is because DP is a nominal value that is much smaller than the actual maximum DPs for raw lignins. For example, the DP for lignin M2 is only 4, yet the MWD for this lignin in Fig. 1 extends to 5000 g/mol. In other words DP, as defined here, is a poor indication of the extents of actual raw lignin MWDs. This does not in any way affect the simulation results because those are based on the MW values, which conform with the full MWDs.
M1[6] - M1 [6] – a space. Response: Thanks.
Comparing DP and MW leads to the conclusion on monomers molecular weight: it is of 396-619 g/mol. For comparison, synapyl alcohol, the heaviest phenylpropane unit forming lignins is of 230 g/mol MW. What are monomers in the manuscript?
Response: The reviewer erroneously uses the nominal DP values to estimate the monomer weights. The actual monomer weights, specified from equations in [1], are given in Table 1, and these values are consistent with the reviewer’s estimate.
Why do you use C12 for monomers stoichiometry description? Syringyl FPU contains 11 C atoms only.
Response: Lignin is certainly not a homogeneous polymer of syringyl. Readers can see this very clearly from the numbers of C, H, and O in Table 1, which express a broad range of elemental compositions. I selected 12 C atoms because it is consistent with accurate monomer weights and also gives good convergence properties in the equations I solve to estimate the H- and O-numbers.
Line 201. andthat for monomer decomposition was raised to 113 kJ/mol because the sensitivity to
Response: Thanks.
Table 2. The ratio of energy activation for monomer rupture (113 kJ/mol) and recombination (209 kJ/mol) is very strange: radical recombination occurs without activation barrier usually.
Response: The reaction processes in lig-FC are not elementary chemical reactions and their rate parameters cannot be estimated from thermochemistry. Each process represents large numbers of specific reactions, some of which result in depolymerization and some that make new refractory connections among monomers. For that reason, activation energies in these processes will be much different than those in individual elementary reactions.
Line 234. GHCs, OXYs abbreviations are not decoded. Response: Thanks. Those acronyms are defined at first usage.
Fig 2, right. Fitted lines are not fitted the experimental points.
Response: Only the solid curve should be compared to the data points. There are some discrepancies for the two lowest lignin MWs but these are not large compared to the uncertainties on the measured oils/H2O/OXYs yields.
Line 561. weights of 215 and 251 g/mol. ??? For rapid extended heating at 530 °C at 0.1 MPa, the
Response: Sorry. This sentence was truncated in a previous draft. It has been fully restored.
Figure 3. Evaluation of (open symbols connected by line segments) simulated lignin partitioning for 5000 °C/s with 5s IRP at 530 °C with yields of () oils and () char reported by Marathe et al. [6] for (left) vacuum and (right) atmospheric pressure. What are empty symbols? I don’t understand, why abscissa is number of Lignins, but not Molecular weight?
Response: I do not understand with the correct symbols did not show up in the reviewer’s review copy because they appear in my draft manuscript. I will make sure they are correct in the published manuscript. The version of this figure in the format requested by the reviewer appears in [1]. I used the format with sample numbers here because it clearly shows how the heavy and light fractions of each lignin triplet are accurately simulated by lig-FC.
Maximum sum of oil and char yields is of 95-97%. And what about H2O, CO2 CO formation (Fig. 8)?
Response: The maximum sum of oils and char is 95 -97 % in Fig. 7 only for temperatures cooler than 400 C. For these temperatures in Fig. 8, the yields of all noncondensable gas species are very low. In general, these measured product distributions close mass balances within +/- 5 wt. %, which is as accurate as has been reported.
Figure 4. Upper-left figure – fitting is not satisfied.
Response: The accompanying text explains the reason for these discrepancies, which are that I do not believe that oils MWDs extend to weights greater than 1500 – 2000 g/mol, because heavily oxygenated compounds like bio-oils give strong non-exclusion effects in GPC. So I truncate the simulated oils MWDs and this shifts the contributions from lighter oils above the measured MWDs. If the measured MWDs were corrected for non-exclusion effects, the agreement would be better.
Figure 5. (Left) correlation is not satisfied. Right-lower fitting is not satisfied. decreasing by 9.3% abs and 27% relative. Simulated results differ only by 0.9% abs or 3% relative. Hence, the model do not predict correctly the obvious decrease of the oxygen content in oils while increasing temperature of pyrolysis. Prediction (simulation) of the char composition is also insufficient.
Response: As explained in the accompanying text, the discrepancies in the Mn values for vacuum oils are due to the truncation in the simulated MWDs described in my previous response. I have no explanation for the discrepancies in the lower curve for 0.1 MPa in the right panel. However, the reviewer jumps to the erroneous conclusion that this discrepancy means that lig-FC cannot depict diminished oils-O as devolatilization proceeds. This tendency is evident in Table 3, where the simulated oils-O becomes smaller for progressively hotter temperatures, albeit not by very much. But I seriously doubt that this is the reason for the discrepancy in the right panel of Fig 5. Remember, lig-FC describes only primary devolatilization and omits oils decomposition chemistry.
Simulating model is not described in the manuscript.
Response: Before this manuscript was sent out for review the Editors asked me to eliminate the section that summarized the theory, which I did. I agree that the current version is impossible to comprehend for readers who have not previously read [1], but I did not prepare it that way.
To conclude, the reviewer do not recommend the manuscript for publication, because the model are not described, and the hypothesis on the mechanism of pyrolysis is not efficient, it does not describe the results satisfactorily.
Response: Thanks for the reviewer’s constructive comments and criticisms. Hopefully, the next version will be more acceptable.

Reviewer 2 Report
This manuscript gave the comprehensive understanding of the network depolymerization mechanism for the rapid primary devolatilization of mineral‐free lignins (Lig-FC). The conclusions have clearly clarified the author`s intention. I think this manuscript is well organized, and it is suitable to be published with minor revise.
1. In the section of “Introduction”, it is difficult for reader to directly understand this manuscript`s significance when it is started with the ‘Lig-FC’, therefore whether it is more logical to introduce the significance and difficulties of Lig-FC.
2. Is the validation database in [1] standard database?
3. In every figure`s pallet, it is more convenient to use symbols instead of words.
Author Response
- I agree but my editor asked me to eliminate virtually the entire Introduction due to duplication with [1].
- That validation database is one that I compiled myself.
- Since there is only one equation in this paper, I do not want to use symbols in the figures that have to be defined in isolation.
Reviewer 3 Report
This paper deal with theoretical discussion about the thermal destruction of the mineral‐free lignin. The subject of paper looks strange (why author don't mark this article as "review" or minin-review? The term "mineral-free" looks confused since lignin as natural compounds contains small number of inorganic chemicals) but the discussion and the simulation of experemental parameters makes a good impression. The manuscript can be accepted after minor revision:
-Each values and parameters in this paper does not contain a confidence interval (±x.x). Author should be give the short comment about the precision of the observed parameters.
-Throughout the article author call one of the destruction product as "oil". How did author separate the "char" product of destruction form "oil" products? I think it should be clarify in details what means oil, char, and gas products within discussed context.
Author Response
- Regarding the uncertainties on the test data, the testing team is in a much better position than I am to estimate the uncertainties, so I would like to let the interested reader consult the test publications, which I cite, for that information.
- The resolution of oils in the tests is a very important point that I consider in detail in [1]. But the editor does not allow me to replicate that information again in this paper.
Round 2
Reviewer 1 Report
The manuscrint 2561780 “Simulating the rapid devolatilization of mineral‐free lignins” is described the hypothesis on mechanism of lignins fast pyrolysis.
Some remarks on the text are listed below.
Fig. 1. The title is not descried the right picture.
Table 1. Labels are not decoded, lignins are not described. Rows DP and MW are not connected to other ones. Are they described initial lignins or chars?
M1[6] - M1 [6] – a space.
Comparing DP and MW leads to the conclusion on monomers molecular weight: it is of 396-619 g/mol. For comparison, synapyl alcohol, the heaviest phenylpropane unit forming lignins is of 230 g/mol MW. What are monomers in the manuscript?
Why do you use C12 for monomers stoichiometry description? Syringyl FPU contains 11 C atoms only.
Line 201. andthat for monomer decomposition was raised to 113 kJ/mol because the sensitivity to
Table 2. The ratio of energy activation for monomer rupture (113 kJ/mol) and recombination (209 kJ/mol) is very strange: radical recombination occurs without activation barrier usually.
Line 234. GHCs, OXYs abbreviations are not decoded.
Fig 2, right. Fitted lines are not fitted the experimental points.
Line 561. weights of 215 and 251 g/mol. ??? For rapid extended heating at 530 °C at 0.1 MPa, the
Figure 3. Evaluation of (open symbols connected by line segments) simulated lignin partitioning for 5000 °C/s with 5s IRP at 530 °C with yields of () oils and (p) char reported by Marathe et al. [6] for (left) vacuum and (right) atmospheric pressure. What are empty symbols? I don’t understand, why abscissa is number of Lignins, but not Molecular weight?
Maximum sum of oil and char yields is of 95-97%. And what about H2O, CO2 CO formation (Fig. 8)?
Figure 4. Upper-left figure – fitting is not satisfied.
Figure 5. (Left) correlation is not satisfied. Right-lower fitting is not satisfied.
Table 3. The same remarks. Mesured oxygen content in oils is of 33.9 – 24.6%, decreasing by 9.3% abs and 27% relative. Simulated results differ only by 0.9% abs or 3% relative. Hence, the model does not predict correctly the obvious decrease of the oxygen content in oils while increasing temperature of pyrolysis. Prediction (simulation) of the char composition is also insufficient.
Simulating model is not described in the manuscript.
To conclude, the reviewer does not recommend the manuscript for publication, because the model are not described, and the hypothesis on the mechanism of pyrolysis is not efficient, it does not describe the results satisfactorily.
Responses to Reviewer No. 1 for Polymers 2561780 by Stephen Niksa
The manuscrint 2561780 “Simulating the rapid devolatilization of mineral‐free lignins” is described the hypothesis on mechanism of lignins fast pyrolysis. Some remarks on the text are listed below.
Fig. 1. The title is not descried the right picture.
Response: Thanks. To the caption I added “…for (left) heavy and (right) light lignin triplets.”
Table 1. Labels are not decoded, lignins are not described. Rows DP and MW are not connected to other ones. Are they described initial lignins or chars?
Response: The lignins in Table 1 are denoted by the indices in the first column. For those from [6], I used the same numbering system as this testing team. The others can be cross-referenced with the properties in this table. I think the reviewer wants to know why the product of the monomer weight in the <MM> column and DP does not equal MW for the raw lignin. That is because DP is a nominal value that is much smaller than the actual maximum DPs for raw lignins. For example, the DP for lignin M2 is only 4, yet the MWD for this lignin in Fig. 1 extends to 5000 g/mol. In other words DP, as defined here, is a poor indication of the extents of actual raw lignin MWDs. This does not in any way affect the simulation results because those are based on the MW values, which conform with the full MWDs.
M1[6] - M1 [6] – a space. Response: Thanks.
Comparing DP and MW leads to the conclusion on monomers molecular weight: it is of 396-619 g/mol. For comparison, synapyl alcohol, the heaviest phenylpropane unit forming lignins is of 230 g/mol MW. What are monomers in the manuscript?
Response: The reviewer erroneously uses the nominal DP values to estimate the monomer weights. The actual monomer weights, specified from equations in [1], are given in Table 1, and these values are consistent with the reviewer’s estimate.
I did not get an answer on my question: What are monomers in the manuscript? You write about structural units (Line 105), and what is the structure? I do not understand what is the nominal DP values, a calculation procedure is not described in the manuscript. If you write about nominal DP values, what are real DP and why it is not given? Why do you assume that other readers will not use the nominal DP values erroneously like I did?
Why do you use C12 for monomers stoichiometry description? Syringyl FPU contains 11 C atoms only.
Response: Lignin is certainly not a homogeneous polymer of syringyl. Readers can see this very clearly from the numbers of C, H, and O in Table 1, which express a broad range of elemental compositions. I selected 12 C atoms because it is consistent with accurate monomer weights and also gives good convergence properties in the equations I solve to estimate the H- and O-numbers.
Lignin is certainly not a homogeneous polymer of syringyl – I agree, but syringyl FPU is the heaviest monomer. I do not know lignin phenylpropene units containing more than 11 C atoms. How did you determine accurate monomer weights?
Line 201. andthat for monomer decomposition was raised to 113 kJ/mol because the sensitivity to
Response: Thanks.
Table 2. The ratio of energy activation for monomer rupture (113 kJ/mol) and recombination (209 kJ/mol) is very strange: radical recombination occurs without activation barrier usually.
Response: The reaction processes in lig-FC are not elementary chemical reactions and their rate parameters cannot be estimated from thermochemistry. Each process represents large numbers of specific reactions, some of which result in depolymerization and some that make new refractory connections among monomers. For that reason, activation energies in these processes will be much different than those in individual elementary reactions.
It will be useful for the manuscript to include literature results described similar relation between activation energies of destruction and recombination.
Line 234. GHCs, OXYs abbreviations are not decoded. Response: Thanks. Those acronyms are defined at first usage.
Fig 2, right. Fitted lines are not fitted the experimental points.
Response: Only the solid curve should be compared to the data points. There are some discrepancies for the two lowest lignin MWs but these are not large compared to the uncertainties on the measured oils/H2O/OXYs yields.
I do not see Figure capture for Fig. 2-7. For Fig.1 the Figure capture is separated from the Figure.
Line 561. weights of 215 and 251 g/mol. ??? For rapid extended heating at 530 °C at 0.1 MPa, the
Response: Sorry. This sentence was truncated in a previous draft. It has been fully restored.
Figure 3. Evaluation of (open symbols connected by line segments) simulated lignin partitioning for 5000 °C/s with 5s IRP at 530 °C with yields of () oils and (p) char reported by Marathe et al. [6] for (left) vacuum and (right) atmospheric pressure. What are empty symbols? I don’t understand, why abscissa is number of Lignins, but not Molecular weight?
Response: I do not understand with the correct symbols did not show up in the reviewer’s review copy because they appear in my draft manuscript. I will make sure they are correct in the published manuscript. The version of this figure in the format requested by the reviewer appears in [1]. I used the format with sample numbers here because it clearly shows how the heavy and light fractions of each lignin triplet are accurately simulated by lig-FC.
Maximum sum of oil and char yields is of 95-97%. And what about H2O, CO2 CO formation (Fig. 8)?
Response: The maximum sum of oils and char is 95 -97 % in Fig. 7 only for temperatures cooler than 400 C. For these temperatures in Fig. 8, the yields of all noncondensable gas species are very low. In general, these measured product distributions close mass balances within +/- 5 wt. %, which is as accurate as has been reported.
Figure 4 (Lines 393-394). Upper-left figure – fitting is not satisfied.
Response: The accompanying text explains the reason for these discrepancies, which are that I do not believe that oils MWDs extend to weights greater than 1500 – 2000 g/mol, because heavily oxygenated compounds like bio-oils give strong non-exclusion effects in GPC. So I truncate the simulated oils MWDs and this shifts the contributions from lighter oils above the measured MWDs. If the measured MWDs were corrected for non-exclusion effects, the agreement would be better.
Fitting (Line (200,200)-(1000,1000) ) is not satisfied and is not based by any theory or ideas. Why is it not line (300,400)-(800,600), for example? This fitting seems better.
Figure 5. (Left) correlation is not satisfied. Right-lower fitting is not satisfied. decreasing by 9.3% abs and 27% relative. Simulated results differ only by 0.9% abs or 3% relative. Hence, the model do not predict correctly the obvious decrease of the oxygen content in oils while increasing temperature of pyrolysis. Prediction (simulation) of the char composition is also insufficient.
Response: As explained in the accompanying text, the discrepancies in the Mn values for vacuum oils are due to the truncation in the simulated MWDs described in my previous response. I have no explanation for the discrepancies in the lower curve for 0.1 MPa in the right panel. However, the reviewer jumps to the erroneous conclusion that this discrepancy means that lig-FC cannot depict diminished oils-O as devolatilization proceeds. This tendency is evident in Table 3, where the simulated oils-O becomes smaller for progressively hotter temperatures, albeit not by very much. But I seriously doubt that this is the reason for the discrepancy in the right panel of Fig 5. Remember, lig-FC describes only primary devolatilization and omits oils decomposition chemistry.
Simulating model is not described in the manuscript.
Response: Before this manuscript was sent out for review the Editors asked me to eliminate the section that summarized the theory, which I did. I agree that the current version is impossible to comprehend for readers who have not previously read [1], but I did not prepare it that way.
To conclude, the reviewer does not recommend the manuscript for publication, because the model are not described, and the hypothesis on the mechanism of pyrolysis is not efficient, it does not describe the results satisfactorily.
Response: Thanks for the reviewer’s constructive comments and criticisms. Hopefully, the next version will be more acceptable.
To conclude.
I did not get an answer on my question: What are monomers in the manuscript? You write about structural units (Line 105), and what is the structure?
How did you determine accurate monomer weights and nominal DP values?
Figure 4 (Lines 393-394). Fitting (Line (200,200)-(1000,1000) ) is not satisfied, i.e. simulation is not satisfied. Line (300,400)-(800,600), for example, fits experiments better.
Round 3
Reviewer 1 Report
Dear Author, I do not mind, the paper may be accepted.
Responses to Comments in the third Round
2. Lignin is certainly not a homogeneous polymer of syringyl – I agree, but syringyl FPU is the heaviest monomer. I do not know lignin phenylpropene units containing more than 11 C atoms. How did you determine accurate monomer weights?
Response: Syringyl is not the heaviest monomer in lignin and it is misleading to insist that monomers are identical to isolated structures when, in actuality, monomers are interconnected and crosslinked in whole lignins. The monomers in this analysis comprise the structural units plus all interconnections. The essential point is that the assigned compositions and MWs for monomers in this analysis are entirely reasonable. The fact that these values are based on 12 or 11 C-atoms is irrelevant.
You may indeed be correct, and it is not significant whether the model is relevant to the studied process.